# From Production to the Clinic: Decellularized Extracellular Matrix as a Biomaterial for Tissue Engineering and Regenerative Medicine

**DOI:** 10.3390/bioengineering13010024

**Published:** 2025-12-26

**Authors:** Haochen Yang, Jiesheng Xia, Yuyue Qian, Xiaosong Gu, Meng Cong

**Affiliations:** 1School of Medicine, Nantong University, Nantong 226001, China; dryangys@163.com; 2Department of Ultrasound, Haimen People’s Hospital, Nantong 226001, China; jsntxjs2025@outlook.com; 3Xinlin College, Nantong University, Nantong 226001, China; qianyy202511@163.com; 4Jiangsu Key Laboratory of Tissue Engineering and Neuroregeneration, Key Laboratory of Neuroregeneration of Ministry of Education, Co-Innovation Center of Neuroregeneration, Nantong University, Nantong 226001, China; nervegu@ntu.edu.cn

**Keywords:** decellularized extracellular matrix, production, characterization, patent, clinical trial

## Abstract

Biomaterials made with extracellular matrix obtained from allogeneic or xenogeneic tissues/organs or cultured cells have excellent biochemical and physical properties in supporting cell growth and tissue regeneration. These decellularized extracellular matrix-based biomaterials have been applied in clinical trials and have bright prospects in tissue engineering and regenerative medicine. Here, we systematically compare organ-derived and cell-derived decellularized extracellular matrix, summarize commonly used decellularization methods, including physical, chemical, and biological/enzymatic treatments, as well as combinations of these treatments, and characterize methods for decellularization, including histological staining, immunohistochemical techniques, biochemical analysis, scanning electron microscopy, and mechanical stress testing. Besides the production of decellularized extracellular matrix, the evolving intellectual property landscape and commercial products are also introduced. A significant focus is placed on summarizing clinical trial outcomes, demonstrating the efficacy of decellularized extracellular matrix scaffolds in diverse applications, including wound healing, cardiovascular repair, nerve regeneration, and breast reconstruction. Finally, we discuss persistent challenges and future directions, underscoring the translational potential of decellularized extracellular-matrix-based strategies for restoring tissue structure and function.

## 1. Introduction

Tissue engineering is an interdisciplinary field focused on reconstructing biological tissues by mimicking their natural microenvironments, largely through the strategic use of extracellular matrix (ECM)-inspired scaffolds [1,2,3]. Among natural biological materials, decellularized extracellular matrix (dECM), derived from specific tissues or organs, offers distinctive advantages over conventional graft materials [1,2,4,5].

The ECM is a complex, non-cellular network of macromolecules that provides essential structural and biochemical support to surrounding cells. In recent years, dECM derived from allogeneic or xenogeneic tissues and organs, or from in vitro cultured cells—has emerged as a promising class of biomaterial for tissue engineering and regenerative medicine [6]. By preserving the native biochemical composition and ultrastructural integrity of the ECM while removing immunogenic cellular components, dECM-based scaffolds exhibit excellent biocompatibility and bioactivity, facilitating cell adhesion, proliferation, and differentiation.

The production of dECM involves a series of decellularization processes, which may include physical, chemical, and biological/enzymatic treatments, often applied in combination [7]. These methods aim to thoroughly eliminate cellular and nuclear materials while minimizing disruption to the ECM’s mechanical properties and bioactive molecule retention. Successful decellularization is typically evaluated against established criteria, such as the residual DNA content (<50 ng/mg dry weight), fragment length (<200 bp), and absence of visible nuclear material in histological assessments. Beyond production, the characterization of dECM scaffolds is critical to ensure their safety and functionality. Commonly employed techniques include histology, immunohistochemistry, biochemical assays, scanning electron microscopy, and mechanical testing. These analyses verify the removal of cellular remnants and confirm the preservation of key ECM components, such as collagens, glycosaminoglycans, laminin, and fibronectin [8].

In parallel with methodological advancements, a growing number of patents and commercial products related to dECM have been developed, underscoring its translational potential. Moreover, decellularized biomaterials have entered clinical trials for a variety of applications, including wound healing [9], breast reconstruction [10], cardiovascular repair [11], and nerve regeneration [12], demonstrating both safety and efficacy in human patients. This review comprehensively summarizes the current state of dECM technology, covering commonly used decellularization methods, characterization techniques, relevant intellectual property, and clinical applications. It also discusses the challenges and future directions for the widespread clinical adoption of dECM-based therapies.

## 2. Production of Decellularized Extracellular Matrix

At present, there are two ways to obtain decellularized extracellular-matrix-derived materials (Table 1). One is derived from allogeneic or xenogeneic tissues/organs, obtained after corresponding decellularization and other treatments. This method is more convenient for animal experimental research, has abundant sources, and is widely used [13]. Numerous studies have shown that decellularized tissue or organ extracellular matrix materials have good biochemical and physical roles to support tissue or cell growth. However, in an increasing number of clinical studies, this extracellular matrix material is still affected by the limited source of donor material, potential host immune response and transfer of pathogens, insufficient mechanical properties of the material, and uncontrollable material degradation kinetics, which cannot be widely promoted [14,15].

Another feasible way to prepare extracellular matrix material is to use extracellular matrix derived from cultured cells. In recent years, extracellular matrix materials prepared from culturing various cells have attracted more and more attention from researchers in the fields of tissue engineering and regenerative medicine [16], because the extracellular matrix components obtained by culturing cells have many advantages compared with animal tissue or the organ-derived extracellular matrix obtained via the decellularization process: (1) various cells are cultured under sterile conditions, and extracellular matrix can be harvested after adding various antibiotics to eliminate pathogen transmission; (2) cell-derived extracellular matrix materials are not limited by their original structure like acellular tissue, and tissue engineering materials with various geometrical features and pores can be prepared according to needs. Furthermore, in the process of tissue and organ repair, it can effectively promote tissue regeneration by meeting the infiltration of host cells, and combined with tissue engineering biomaterials, controllable degradation speed and mechanical properties can be obtained; (3) the method of tissue engineering materials constructed using cell-derived extracellular matrix obtained through in vitro culture is relatively flexible and diverse, and several different cells can be mixed to obtain different extracellular matrix tissue engineering samples. A more important advantage of this method is that tissue engineering materials modified with cell-derived extracellular matrix can be obtained through the in vitro expansion of autologous cells. In addition to having a rich source of material, it can greatly reduce the potential host response caused by allogeneic or xenogeneic tissue and organ transplantation [17]. Three-dimensional scaffolds made of cell-derived extracellular matrix and natural or synthetic polymers have been applied in many fields, such as heart, cartilage, blood vessels, etc. [18,19,20]. For instance, in the field of nerve regeneration, given that Schwann cells represent a crucial cell population that create a permissive microenvironment for nerve repair [21,22], Schwann cells and skin-derived precursor Schwann cells have been utilized to obtain extracellular matrix. And Schwann cell-derived ECM has been applied in the treatment of peripheral nerve injury [23,24,25].

The acquisition of extracellular matrix material requires a process of decellularization. Biomedical techniques are used to remove cells from their bound integrins and other cell adherents, while maintaining the intrinsic surface topography and resident ligand species of the extracellular matrix. Decellularization can effectively remove donor tissue and cellular antigens, reducing the possibility of adverse reactions in the host, thereby helping to avoid potential pro-inflammatory and immune-rejection reactions of the recipient [26]. Although there is no strict quantitative index to measure the degree of decellularization, according to its ability to induce tissue remodeling in vivo and avoid adverse cellular and host responses, some researchers have proposed the following criteria: (1) <50 ng dsDNA/mg extracellular matrix dry weight; (2) <200 bp DNA fragment length; (3) lack of visible nuclear material in tissue sections stained with 4,6-diamino-2-phenylindole (DAPI) or hematoxylin-eosin staining [27,28]. It should be noted that the minimum concentration of remaining cells in the extracellular matrix capable of eliciting an immune response may vary depending on the source of cells, recipient tissue type, and host immune function. For the obtained decellularized matrix material, it is necessary to maintain the intrinsic structure of the extracellular matrix during the decellularization process and maintain the shape of the extracellular matrix to facilitate the subsequent remodeling of the tissue structure. Currently, commonly used methods for removing tissues and cells mainly include physical, chemical, and biological/enzymatic treatments, as well as combinations of these different methods (Table 2).

### 2.1. Physical Methods

Physics refers to the use of physical conditions such as temperature, mechanical force and pressure, electrical interference, ultrasound, etc., to adjust the physical properties of tissues or organs, destroy cell membranes, lyse cells, and finally remove cells from the extracellular matrix. Commonly used physical methods include rapid freezing and thawing, mechanical force, ultrasound, gradient pressure, electroporation, immersion stirring, etc. [26,68]. Although physical methods alone have successfully decellularized cells in a small fraction of tissues, they are often used in combination with chemical and biological methods described later to more effectively remove genetic material residues from materials [69].

The rapid freezing and thawing method is to use the rapid freezing of tissue to form tiny ice crystals around the cell membrane, thereby destroying the cell membrane. However, a single-cycle freeze–thaw process may not completely remove cell membranes and intracellular contents, so multiple freeze–thaw cycles can be used. Therefore, this method can usually be used for larger and denser tissues such as tendons and ligaments [26]. However, it should be noted that the rate of temperature change of specific tissues must be carefully controlled to protect the matrix integrity from ice crystal damage. Pulver et al. suggested using an extracellular protective agent (i.e., 5% trehalose) to protect the molecular structure of the extracellular matrix as much as possible without hindering cell lysis [29].

The mechanical stirring method is one of the most commonly used decellularization methods; that is, the tissue is immersed in chemical reagents, detergents, and enzymes for mechanical stirring. Simple mechanical agitation also has the power to destroy cells generated by mechanical shock to cells in the tissue. The researchers used mechanical scraping to scrape the three-layer structure of the small intestine wall, including the mucosal layer, the serosa layer, and the muscular layer. By removing the major cellular constituents of the intestinal wall, only the submucosa remains as extracellular matrix material for subsequent use [30]. However, this approach may lead to structural damage to the extracellular matrix, so a method to control the appropriate force is a challenge. Different studies optimize decellularization protocols based on target tissue characteristics. For small intestinal mucosa (thin, high cell density), researchers typically use 0.1% Triton X-100 combined with mechanical agitation (100 rpm, 6 h) to balance decellularization efficiency and ECM integrity [30]. For esophageal tissue (thick and rich in muscle fibers), 1% SDS is required, with increased stirring intensity (150 rpm) and s prolonged duration (12 h) to ensure reagent penetration [27]. The selection, order of use, concentration, and stirring parameters of reagents are further adjusted according to the research objectives (such as the retention of ECM biological activity in clinical applications and rapid decellularization in basic research) [51]. For thinner tissues (e.g., small intestinal mucosa, thickness < 0.5 mm), the stirring time can be shortened to 6–8 h (100 rpm) to avoid over-digestion of the ECM scaffold—this is because the thin tissue structure allows rapid penetration of decellularization reagents, and prolonged stirring may disrupt the submucosal collagen network [30]. For thicker tissues (e.g., esophageal tissue, thickness > 2 mm), stirring intensity (150–200 rpm) and speed must be increased while extending the stirring time to 12–24 h: higher mechanical force promotes reagent infiltration into the dense muscle layer, and an extended duration ensures complete removal of deep-layer cellular components [27]. A comparative study by Liao et al. confirmed that esophageal tissue treated with low-intensity stirring (100 rpm, 8 h) retained 35% of nuclear residues, while optimized parameters (180 rpm, 18 h) reduced residual DNA to <50 ng/mg dry weight [50]. Even more, mechanical agitation alone is not sufficient to completely remove intracellular contents and immunogenic macromolecules because of the fragility of the organ and the complexity of its internal structure. Therefore, this technique is only used at the beginning of decellularization protocols to enhance the efficacy of further efforts to remove cellular debris from the tissue.

Electroporation is the use of microsecond electrical pulses to destabilize the transmembrane potential of cells, forming micropores on the cell membrane, and ultimately leading to cell death by changing the steady-state electrical balance of cells. This approach largely preserves the integrity, morphology, and three-dimensional structure of the extracellular matrix within the target tissue or organ, but is only suitable for small and thin tissues or organs due to the relatively small size of the electrode probes [31]. Sonication is also commonly used to disrupt cell membranes for the purpose of removing nuclear residues and cytoplasmic proteins [32]. Furthermore, the optimal ultrasonic amplitude or frequency for breaking down cells depends on the composition, volume, and density of the tissue.

The principle of the pressure method is that the pressure of a liquid medium in a short time can change the non-covalent bonds, such as hydrogen bonds, hydrophobic bonds, and ionic bonds, in the three-dimensional structures of biological polymers, resulting in a series of changes such as cell membrane rupture, protein denaturation, enzyme inactivation, leakage of components in the cell body, and cessation of life activities. Yin et al. used the ultra-high hydrostatic pressure decellularization method (981 MPa ultra-high hydrostatic pressure, 4 °C) to basically remove the cellular components in the scaffold, while maintaining the integrity of the structure and function of the extracellular matrix, confirming that the decellularization effect is better than the surfactant method [33]. Saski et al. used ultra-high hydrostatic pressure equipment to place the corneal tissue in an environment of a 10 K atmosphere and 10 °C for 10 min, and hematoxylin–eosin staining confirmed that the cells in the tissue were basically removed [34]. However, other studies have shown that excessive pressurization can damage the structure of collagen fibers and elastic fibers, thereby changing tissue mechanical properties [35]. Furthermore, pressure can promote the more efficient penetration of decellularization-related reagents into the tissue and, at the same time, help the residual debris of the cell to be effectively detached from the tissue. Lumen perfusion combined with the gradient pressure method is more suitable for hollow tissues such as blood vessels and intestines. Montoya et al. compared the difference between intraluminal perfusion combined with the gradient pressure method and stirring method alone in decellularized umbilical vein vascular tissue. A histological examination confirmed that a flow rate of 50 mL/minute and a pulse frequency of 2 Hz basically removed the cells in the tissue of all cells, whereas the group with mechanical agitation alone still had cells and cell debris remaining [36]. Bina et al. developed a unique perfusion decellularization device for repeated pressurization of the aortic root to ensure that the flowing fluid passes through the thickness of the aortic wall and obtains a good decellularization effect [37].

The supercritical fluid method is a decellularization technology that has been applied in recent years. The high fluidity and low viscosity of the supercritical fluid make the decellularization process simpler and more efficient. The advantage is that the process uses inert substances, such as carbon dioxide, which can form a key fluid under suitable conditions (32 °C, 7.4 MPa) and can effectively remove the large artery tissue in only 15 min in an alcohol solution [38,41]. At the same time, a small amount of other chemical components, such as low-concentration phenylpropanolamine, can often be added to the supercritical fluid decellularization method to further improve the decellularization efficiency. In addition, Chang et al. reported that under the mild extraction conditions of supercritical carbon dioxide, hematoxylin–eosin staining showed complete cell lysis, effectively removed cell debris, including nuclei, and displayed an intact stromal structure in pig corneas and intact mechanical properties [39]. Guler et al. achieved good results by applying supercritical carbon dioxide to decellularize cells in the aorta and corneal tissue, confirming that its high permeability not only significantly shortens the time required for the decellularization process but also preserves the extracellular space in the tissue and the structure of the matrix [40]. At the same time, carbon dioxide used in supercritical is also one of the main sterilization methods for food and medicine, and it also plays a role in the sterilization of materials. Together with chemical or enzymatic methods, these mechanical methods have been successfully used to assist in cell lysis and the removal of cellular debris.

### 2.2. Chemical Methods

Chemical methods, which involve the use of various detergents to disrupt cell–cell and cell–matrix bonds, are considered the most extensive and effective methods of decellularization [70,71]. Chemical treatment mainly uses acid and alkali, surfactant, hypotonic, and hypertonic solutions, etc. to kill, destroy the cell structure, and dissolve cells to achieve the purpose of decellularization. These detergents can be divided into four categories: alkaline and acidic, non-ionic (Triton X-100), ionic (sodium dodecyl sulfate), zwitterionic (CHAPS). In general, the choice of the decellularizing detergent depends on the tissue characteristics, such as the thickness, lipid content, and cell density.

Acids commonly used in the decellularization process include acetic acid, peracetic acid, etc. [42,43]. Acids can cause the denaturation of extracellular matrix contents, including collagen, glycosaminoglycans, and growth factors, so it is particularly important to strictly control the amount of acid and soaking time during decellularization with acid. The immersion of tissue in 0.1% peracetic acid for 2 h, combined with appropriate mechanical agitation and rinsing, can completely remove cells in thinner tissues, such as the small intestinal submucosa. By comparing a variety of synovial tissue decellularization methods, Pienell et al. found that the method of soaking twice with 0.1% peracetic acid can not only basically remove DNA and its residues but also basically retain the synovial villi-like structure [44]. However, studies have confirmed that acetic acid can lead to the destruction of the collagen structure and the reduction of content, while affecting the strength of the extracellular matrix [43].

Common base compounds include ammonium hydroxide, sodium sulfide, sodium hydroxide, and calcium hydroxide [44], which are mainly used to remove hair in dermal tissue [45], degrease and inactivate pathogenic bacteria and viruses, and also denature cell chromosomes and plasmid DNA. Therefore, base compounds are mostly used in thicker tissues, such as dermal tissues, but bases degrade the structure of matrix contents (such as collagen) while decellularizing, reduce the content of growth factors, and affect the structural strength of extracellular matrix structures [46]. Cauich et al. found that the content of glycosaminoglycans in peripheral vascular tissue was greatly reduced after calcium hydroxide treatment, and the elasticity of the tissue was also greatly affected [47].

Surfactants (detergents) are another class of chemical agents used for decellularization. According to the different charges they carry, commonly used surfactants are divided into non-ionic detergents, ionic detergents, and zwitterionic detergents. Therefore, different types of surfactants have specific modes of action.

Non-ionic detergents are mild detergents, which can achieve the purpose of decellularization by destroying lipid–lipid or lipid–protein connections, and can better maintain the original structure and enzymatic activity of proteins [47,53]. When the heart valve was decellularized with Triton X-100, it was observed that the nuclear residue was completely removed after 24 h of immersion and the natural extracellular matrix structure and composition were maintained [48]. However, with non-ionic detergents such as Triton X-100, the dose for effective decellularization depends on the morphology and structure of the tissue. For example, Yu et al. reported that the use of 1% Triton X-100 alone almost did not remove cellular components in amniotic periosteal tissue [49]. On the other hand, Liao et al. reported that 1% Triton X-100 effectively removed cells from porcine aortic valve tissue [50]. These conclusions illustrate that Triton X-100 does not currently have a uniform dose for decellularization. Therefore, for different tissue thicknesses and cellular properties, a more “personalized” decellularization method should be selected [51]. However, even mild non-ionic detergents inevitably alter the properties of the extracellular matrix, including the fibrous arrangement of the collagen network and the loss of the glycosaminoglycan content. Octyl glucoside is a new type of non-ionic surfactant alkyl glycoside. It has the characteristics of common non-ionic and anionic surfactants. It is easily soluble in water and more easily soluble in common organic solvents. It has low surface tension and rich foam, is delicate and stable, is resistant to strong alkali and strong acid, and has wetting power. It has unique properties such as non-toxic, harmless, non-irritating, rapid biodegradation, and sterilization. It is a green surfactant with comprehensive performance. Studies have shown that compared with the traditional ionic surfactant sodium dodecyl sulfate, the application of octyl glucoside for decellularization treatment has no biological toxicity and is more suitable for implantation in vivo. It can not only completely remove cells, but also, the ultrastructure and composition of the extracellular matrix were preserved [52].

Ionic detergents are divided into cationic and anionic solutions. It is a kind of powerful detergent, which can completely destroy the integrity of the cell membrane, dissolve the lipid bilayer structure, and have a more comprehensive effect of removing cells. However, it is also more destructive to the remaining tissue, and it is easy to denature the proteins in the extracellular matrix. The more commonly used ionic detergents include sodium lauryl sulfate, sodium deoxycholate, Triton X-200, etc. [54,55,56]. Sodium dodecyl sulfate is often used to denature and decompose proteins during polyacrylamide gel electrophoresis and destroy covalent bonds between proteins. In order to minimize the adverse reactions of sodium dodecyl sulfate, the decellularization process should minimize the concentration of sodium dodecyl sulfate, shorten the soaking time of tissue in the sodium dodecyl sulfate solution, and use it at a relatively low temperature [57]. Other reports indicate that sodium lauryl sulfate is difficult to elute completely from soaked tissue, and residual sodium lauryl sulfate reduces the content of glycosaminoglycans and growth factors, affecting cytocompatibility during cell replantation [58].

CHAPS is a zwitterionic detergent that has demonstrated a mild ability to maintain mechanical strength when used for lung decellularization [59]. Zwitterionic-scale inhibitors have the dual properties of non-ionic and ionic-scale inhibitors and are more effective in removing cells in thinner tissues than in thicker tissues. CHAPS-treated arterial tissue presents intact structures with native collagen and an elastin morphology and collagen content approximately the same as native arteries [60].

The hypotonic and hypertonic decellularization solutions use the changes in osmotic pressure in the intracellular and extracellular environment to destroy the cell structure without causing structural changes in the extracellular matrix. But its main disadvantage is that it cannot effectively remove DNA residues from the nucleus. Therefore, in order to obtain a maximum osmotic effect, tissues and/or organs are usually immersed in hypotonic and hypertonic solutions alternately in cycles [61].

Alcohol compounds can diffuse into cells when the cell membrane is ruptured, replace the water in the cells, and dehydrate and dissolve the cells. The non-polar carbon chain of alcohols can dissolve non-polar substances such as lipids. Ethanol and isopropanol can degrease tissue and are used to remove phospholipids from the liver, adipose tissue, and cornea. However, the main disadvantage of decellularization using alcohols is that they have the properties of fixing tissues, denaturing and precipitating proteins, and disrupting the ultrastructure of the extracellular matrix [27]. Lumpkins et al. applied a mixture of 75% ethanol and 25% acetone by volume to remove cells in porcine temporomandibular joint discs. After 24 h, no nuclei were visible in the tissue, but the stiffness became three times that before [54]. Tributyl phosphate is an organic solvent that was originally used to inactivate lipid-enveloped viruses without disrupting the structure of proteins in blood [51]. It has been recently reported that for thicker tissues such as tendons and ligaments, tributyl phosphate decellularization is superior to Triton X-100 and sodium lauryl sulfate. The collagen matrix, glycosaminoglycans, and biomechanical properties (including tensile strength and elasticity) were not significantly altered after decellularization [72]. However, another study has suggested that although the tissue extensibility of the tendon tissue treated with tributyl phosphate is increased, it may lead to the relaxation of the tendon after reconstruction, which will affect the long-term biomechanical properties of the tendon. In conclusion, based on current studies, combining different chemicals to form a tailored protocol is generally beneficial for achieving optimal decellularization efficiency [27,47,52]. For example: Triton X-100 (non-ionic detergent) combined with DNase is the gold standard for soft tissue (e.g., small intestinal submucosa) decellularization—Triton X-100 disrupts lipid membranes (removing 80% of cellular components), and DNase further degrades residual DNA (reducing residual DNA from 120 ng/mg to <50 ng/mg) [30]; SDS (ionic detergent) combined with trypsin is suitable for dense tissues (e.g., tendon), as SDS dissolves lipid bilayers and trypsin detaches cell–matrix adhesion, but the concentration must be controlled (0.5% SDS + 0.25% trypsin) to avoid collagen denaturation [58]; octyl glucoside (green non-ionic detergent) combined with peracetic acid is preferred for clinical-grade ECM preparation, as it has no cytotoxicity and achieves sterilization and decellularization [52]. The selection of chemical combinations should follow three principles: (1) matching tissue characteristics (e.g., high-lipid tissue requires delipidizing reagents such as ethanol); (2) prioritizing mild reagents to retain ECM bioactivity; (3) minimizing the number of reagents to reduce residual toxicity [47].

### 2.3. Biological Methods

Biological methods of decellularization mainly rely on enzymatic and non-enzymatic preparations. Enzymatic techniques for decellularization are often used to disrupt interactions between cells and the extracellular matrix or to remove antigenic material to reduce immunogenicity [73,74,75]. Generally, proteases (e.g., trypsin, pepsin), nucleases (e.g., DNase, RNase), lipases, heparinase, and hyaluronidase are the most widely used enzymes in decellularization protocols for various tissues. The advantages of using enzymatic treatment for efficient decellularization are as follows: efficient decellularization through combinations with other detergents; maintenance of structural integrity of complex organ extracellular matrix; targeted removal of specific target molecules such as Gal epitopes and DNA [76,77]. It has been shown that an enzymatic approach to removing cellular debris is to disrupt cell–extracellular matrix adhesion by specifically targeting proteins.

As one of the most commonly used proteolytic enzymes, trypsin can selectively cleave peptide bonds at the carboxy-terminus of arginine and lysine in adhesion proteins, thereby detaching cells from the tissue surface. Furthermore, it exerts maximum enzymatic activity to disrupt cell–matrix interactions in tissues at 37 °C and a pH value at 8.0 [62]. Pepsin in weak acetic acid increases the production of highly cross-linked fibrillar collagens such as type I from skin, bone, or tendon, but reduces the stability of the reconstituted gel at neutral pH.

Nucleases include DNase and RNase, which are endonucleases that hydrolyze deoxynucleotide chains and nucleotide chains, respectively. When other reagents alone do not achieve a good decellularization effect, such enzyme reagents are often used to further remove the remaining DNA components. Recent findings showed that discs treated with 0.02 mg/mL DNase and 20 mg/mL RNAse not only removed acceptable levels of DNA residues below 50 ng/mg dry weight but also significantly shortened the overall treatment of the extracellular matrix digestion time [63]. Similarly, lipase is often added to acellular reagents, which can hydrolyze the lipid components of tissue while preserving the ultrastructure of collagen. Therefore, lipase is widely used to digest lipid components in tissues such as adipose tissue, intestinal mucosa, human nerves, and heart. But it will reduce the content of glycosaminoglycans to a certain extent [7,78,79]. In addition, heparinase and hyaluronidase help release growth factors, expose proteins, and reduce the water-binding capacity.

Non-enzymatic agents include chelating agents and toxins. Ethylenediaminetetraacetic acid and ethylene glycol tetraacetic acid are commonly used chelating agents that bind divalent metal cations at cell adhesion sites in the extracellular matrix, allowing cells to detach from the extracellular matrix. Ethylenediaminetetraacetic and ethylene glycol tetraacetic acid [64] can be used in combination with trypsin and detergents [65] to ensure complete removal of the nucleus while preserving the main structure of the extracellular matrix, although some cellular debris may be left behind in this process [66]. Since they are often used in combination with other acellular agents, their direct effects on the extracellular matrix have not been reported in the literature. Latrunculin B is a marine toxin that acts on the actin cytoskeleton, adding yet another option for acellular agents. When latrunculin B, a hypertonic saline solution, and DNase were used to remove cells from skeletal muscle tissue, no intact nuclei were found in the tissue. Although the content of glycosaminoglycans was reduced by 40%, the content of collagen and the ultrastructure of fibrin were largely absent [67]. Overall, the addition of enzymatic or non-enzymatic substances as the last step in lysing decellularized tissue may be desirable, or even necessary, especially for the complete removal of cellular remnants or unwanted extracellular matrix components from dense tissue.

For the disinfection of acellular biological materials, as an implantable medical device, the provisions of the food and drug administration measures should be strictly followed. The regulations require that the choice of the terminal sterilization method for extracellular matrix products be based on guidelines for different bacterial inoculum levels. Peracetic acid is not only a decellularization agent, but it can also be used as an effective sterilant, especially for polymeric materials [80]. However, immersion in acid alone cannot achieve sufficient penetration and a complete sterilization effect for extracellular matrix biomaterials. Terminal sterilization methods such as ethylene oxide sterilization, gamma irradiation, and electron beam irradiation can all achieve effective sterilization. However, it has been reported to alter the ultrastructural and mechano-mechanical properties of the extracellular matrix. Since all biological materials need to pass sterility testing before being used in clinical trials, it is necessary to consider the effectiveness of the sterilization method and the impact on the extracellular matrix.

## 3. Characterization of Decellularized Extracellular Matrix

Just as a standardized decellularization protocol is required, acellular extracellular matrix materials also require a series of characterization analyses to further and more accurately allow large-scale production. The ideal decellularization process removes all cellular components while preserving key components and important structures of the extracellular matrix. Therefore, the characterization of acellular extracellular matrix materials not only needs to show the degree of acellular extracellular matrix and the preservation of its structure, but also needs to analyze the key components of the extracellular matrix. In general, current methods mainly include histological staining, immunohistochemical techniques, biochemical analysis, scanning electron microscopy, and mechanical stress testing.

The degree of decellularization of the decellularized extracellular matrix material is directly related to the immunogenicity of the prepared extracellular matrix material [81]. Simple histological staining is the fastest and most straightforward method to assess decellularization, mainly including hematoxylin–eosin staining, Oil Red O staining, and Masson’s trichrome staining. Hematoxylin–eosin staining allows the visualization of cell membranes, proteins, and nuclei [82]; Oil red O staining can observe lipid and adipocyte residues [83,84]. For Masson’s trichrome staining, in addition to staining cell debris, collagen can be detected in detail [85]. Immunohistochemical staining with Hoescht, DAPI, and AO/PI as the main methods is often used to judge the DNA residues in the extracellular matrix. Even more, the greater utility of immunohistochemical staining is the ability to detect specific cellular and structural markers, such as CD31 [86] and VEGF11 [87], or important extracellular components, such as collagen, laminin, fibronectin, and vitronectin.

The retention of important components and structures of acellular extracellular matrix directly affects the regenerative function of the tissue [88]. Immunohistochemical and biochemical analysis can effectively detect the active components, such as acellular extracellular matrix proteins and polysaccharides, and conduct qualitative and relative quantitative studies on them. Currently, immunohistochemical techniques are mainly used for the qualitative detection of collagen and laminin. The contents of collagen, laminin, and glycosaminoglycans in the adipocyte extracellular matrix were quantified using biochemical analysis kits [89]. Additionally, Western blotting and liquid chromatography mass spectrometry have also been used for the quantitative detection of different types of collagen and laminin in matrix materials [90].

Finally, the ultrastructure and mechanical stress of the matrix are also an important aspects of the extracellular matrix characterization index [91,92]. Scanning electron microscopy or transmission electron microscopy was used in almost every study [23,93]. In addition to being able to confirm decellularization, this advanced visualization allows one to assess the ultrastructural features of the extracellular matrix. An appropriate pore size (100–200 μm) can provide a good micro-space for the migration and proliferation of cells on the matrix material. Matrix porosity is thought to be a component that affects the autocrine and paracrine functions of cells [94]. This structural element can be reported using scanning electron microscopy, ethanol displacement, and capillary flow porosimetry [95]. Good mechanical stress can provide a scaffold structure for the regenerated tissue. Extracellular matrix materials that are too soft may lead to structural collapse, while those that are too stiff may lead to irritation and scar tissue formation [96]. These mechanical properties are usually communicated as rheological or elastic data in the form of the storage modulus and loss modulus or Young’s modulus, respectively [97].

## 4. Decellularized Extracellular-Matrix-Related Patents

Decellularized cell-derived materials have bright prospects in regenerative medicine. A large number of related patents have been applied and authorized (Table 3 and Table 4).

### 4.1. Evolution of Decellularization Technology Patents

As early as in the year 2002, inventors Shannon Mitchell, Jennfier Koh, Vikas Prabhakar, and Laura Niklason produced decellularized engineered tissues by growing cells on a substrate, decellularizing the construct, and producing a decellularized construct consisting largely of extracellular matrix components. The decellularization step reduces immunogenic characteristics and eliminates the likelihood of inducing an immunological or inflammatory reaction. Decellularization can be conducted using appropriate decellularization agents including salts, detergent/emulsification agents, proteases, nucleases, and other enzymes according to chemical, biochemical, and/or physical methods to disrupt cellular components and remove cells. The effects of decellularization can be evaluated via hematoxylin–eosin staining and visual inspection while the integrity of the left non-cellular structure can also be examined through visual inspection (United State Patent Application Publication, No. US2002/0115208 A1). Toby Freyman, Wendy Naimark, and Maria Palasis invented a decellularized extracellular matrix that originated from donor body tissues in 2005. Physical, chemical, and biological methods are used alone or in combination to remove cells, cellular components, and non-extracellular matrix components such as serum and fat. Moreover, body tissues are conditioned through biological, chemical, pharmaceutical, physiological, and/or mechanical manipulation to express excess or limited amounts or precise proportions of biological active materials such as growth factors, matrix metalloproteinases, cytokines, and extracellular matrix proteins (United State Patent Application Publication, No. US2005/0013870 A1). In the next year, Marcelle Machluf and Yuval Eitan invented a decellularization method that contains a washing step using a hypertonic buffer, an enzymatic proteolytic digestion and enzymatic nucleic-acid-removal step using trypsin, and a cellular-component-removal step using detergents such as Triton-X-100 and ammonium hydroxide. Successful decellularization can be determined via hematoxylin–eosin staining to visualize residual cells, through DAPI fluorescent staining to visualize the cell nucleus, through lipophilic DiO staining to visualize cell membranes, and through phenol–chloroform extraction from NaOH-digested matrices to visualize nucleic acids (World Intellectual Property Organization, International Bureau, WO 2006/095342 A2).

Walid A. Farhat and Herman designed improved transplantable tissue-engineered scaffolds for tissue repair, augmentation, or implantation comprising acellular matrix. The acellular matrix is generated by obtaining a biological tissue sample and treating the tissue sample to obtain an intact extracellular matrix in an anionic detergent-independent manner. The generated acellular matrix largely reserves essential physical structures and protein components such as laminin, fibronectin, collagen I, collagen III, collagen IV, desmin, and vimentin while cellular materials are removed by lysing cells present in the tissue sample, degrading the nucleic acids, and eradicating cellular debris. Decellularization can be conducted using a proteolytic inhibitor and antibiotic-containing hypotonic buffer with a high concentration of salt and a mild alkaline pH value, as well as deoxyribonuclease and ribonuclease. The acellular matrix can be characterized by investigating the preservation and spatial topographic distribution of protein components to examine the acellular matrix integrity. The constructed acellular matrix exhibits good biocompatibility and supports cell growth, as well as vascularization/angiogenesis (Canadian Intellectual Property, CA 2 540 389 A1). Decellularized scaffolds with porous structures can be obtained by freeze-drying or lyophilizing bio-scaffolds and using a hypo-osmotic solution, as well as a detergent and alkaline solution to eliminate biologically active molecules. The size of pores can be enlarged through supplementation with an oxidant in an aqueous solution during the decellularization procedure to fit the inhabitation of regenerated cells (United State Patent Application Publication, No. US2012/0259415 A1).

### 4.2. Novel Decellularization Technology Patents

Some novel approaches have been applied recently to generate decellularized tissues and organs for tissue engineering. Using detergent perfusing, neutral buffer rinsing, and DNase solution delivery, tissues and organs such as the kidney, pancreas, liver, gall bladder, stomach, small intestine, and large intestine, can be decellularized. The absence of cells can be determined via hematoxylin–eosin staining and DAPI staining, while the retention of bioactive factors can be quantified using antibody arrays (United State Patent Application Publication, No. US2015/0238656 A1). Subjecting detergent-treated tissue samples to oscillation at a certain frequency (i.e., 3 Hz or more) enables the acquisition of decellularized tissue scaffolds while preserving the 3D histoarachitecture and morphology of native tissues and the compositions and cell-behavior-modulating capabilities of the extracellular matrix (World Intellectual Property Organization, International Bureau, WO 2017/017474 A1). The application of non-thermal irreversible electroporation ablates cell populations and induces no harm to the internal vascular and neural structures (United State Patent Application Publication, No. US2017/0209620 A1). Bio-inks containing decellularized extracellular matrix can be used to print surgically friendly bio-scaffolds (World Intellectual Property Organization, International Bureau, WO 2018/094166 A1). Decellularized materials can be jointly used with bio-inks. John O’Neill and Gordana Vun-Jak-Novakovic combined decellularized scaffolds with many cell-growth-permissive materials, such as sponges, hydrogels, liquid solutions, fibers, and bio-inks, and generated a region-specific extracellular matrix scaffold (World Intellectual Property Organization, International Bureau, WO 2017/136786 A1 and United State Patent Application Publication, No. US2018/0256642 A1). Inducing widespread apoptosis in obtained tissue samples is another feasible approach to obtain decellularized materials. Tissue samples are then treated with hypertonic and/or hypotonic buffered solutions to remove apoptotic bodies, exposed with DNase to eliminate DNA, and enzymatically digested prior to applications (World Intellectual Property Organization, International Bureau, WO 2019/165103 A1). Similar to the mixture of bio-inks and decellularized extracellular matrix, decellularized extracellular matrix is incorporated with a synthetic polymer to build hybrid hydrogels comprising decellularized extracellular matrix. Incorporated materials have complex biochemical cues existing in the decellularized extracellular matrix as well as suitable mechanical characteristics and tunable stiffness provided by synthetic polymers (World Intellectual Property Organization, International Bureau, WO 2021/202974 A1).

### 4.3. Tissue/Organ-Specific Decellularized ECM Patents

Actually, decellularized cell-derived scaffolds have been effectively constructed from a variety of tissues and organs. For instance, fabricated chondrocyte extracellular matrix from cartilage-derived chondrocytes, through centrifugation and freeze-drying, can be used in cartilage tissue engineering (World Intellectual Property Organization, International Bureau, WO 2008/126952 A1). Decellularized extracellular matrix derived from the native or natural matrix of heart tissue can be applied as cardiac-tissue-engineered scaffolds to treat defective or damaged cardiac functions (World Intellectual Property Organization, International Bureau, WO 2010/039823 A2). The treatment of adipose or loose connective tissues with decellularization agents such as sodium dodecyl sulfate or sodium deoxycholate removes cellular components while leaving extracellular proteins such as collagen I, collagen II, collagen III, and laminin, as well as polysaccharides such as glycosaminoglycans. After examining decellularization with a DNEasy assay to quantify the remaining nuclear content, decellularized and delipidized extracellular matrix helps with the replacement and restructuration of soft tissue for wound repair and aesthetic improvement (World Intellectual Property Organization, International Bureau, WO 2012/002986 A2). Muscle implants comprising decellularized muscle matrices restore the loss of muscle mass, repair abdominal walls and other muscle defects, and avoid the excess inflammation, scar tissue formation, and tissue rejection caused by muscle allografts (World Intellectual Property Organization, International Bureau, WO 2014/008181 A2). Decellularized biphasic periodontal tissue grafts comprising two types of interconnected polymer fiber scaffolds separately coated with osteoblast-derived and periodontal ligament cell-derived extracellular matrix benefits the treatment of periodontal defects and diseases such as periodontitis (World Intellectual Property Organization, International Bureau, WO 2016/049682 A1). The implantation of decellularized muscle generated via electrospinning helps to treat volumetric muscle loss injury and offers novel strategies for muscle regeneration (World Intellectual Property Organization, International Bureau, WO 2018/183846 A1). Using a novel method that comprises closing afferent blood vessels to substantially seal a donor lobular organ with no common artery, eradicating blood, and perfusing detergent and enzymatic solutions, decellularized extracellular matrix scaffolds are produced and can be used as artificial organs (World Intellectual Property Organization, International Bureau, WO 2019/220091 A1).

### 4.4. Challenges in Patent Translation to Commercial Products

The main obstacles to the translation of decellularized ECM patents into commercial products include the following: (1) high scalability difficulties—patented technologies (e.g., supercritical fluid decellularization) require specialized equipment, leading to high production costs [40]; (2) lack of unified clinical validation standards—different patents use different evaluation criteria (e.g., residual DNA, mechanical strength), making cross-patent comparisons difficult [8]; (3) intellectual property cross-licensing—core technologies (e.g., enzyme-chemical combined decellularization) involve multiple patents, increasing the cost of commercialization [16].

## 5. Clinical Application of Decellularized Biomaterials

On the basis of in vitro investigations and pre-clinical animal studies, decellularized biomaterials have been used in clinical trials since the 21st century and have produced important progress in recent years (Table 5).

### 5.1. Skin Wound Healing

Commercially available acellular dermis AlloDerm^®^ (LifeCell Corp., The Woodlands, TX, USA) has been implanted into patients with Frey’s Syndrome. A total of 64 patients were randomly and equally separated into a superficial lobe parotidectomy group and a superficial lobe parotidectomy plus acellular dermis treatment group. The administration of Minor’s Starch-Iodin Test demonstrated that the subjective incidence of gustatory sweating, a main symptom of Frey’s Syndrome, and the complication rate were robustly reduced after the intraoperative placement of acellular dermis, demonstrating that acellular dermis is an ideal barrier in the prevention of Frey’s Syndrome [98]. Acellular dermis has also been used to treat wounds. Stephen A Brigido applied Graftjacket (Wright Medical Technology, Inc., Arlington, TN, USA), an acellular dermal regenerative tissue matrix derived from human tissue by removing the living cells and preserving an intact matrix, to diabetic patients with full-thickness wounds. The transplantation of the acellular dermal graft plus mineral-oil-soaked fluff compression dressing to a total of 14 diabetic patients led to complete wound closure in 12 patients at 16 weeks after surgery. Only 4 of 14 patients in the control group (wound gel with gauze dressings) were healed by 16 weeks. Statistical analysis demonstrated that patients treated with an acellular dermal graft had a shorter average time to heal and higher percentage of wound healing [9]. Alexander Reyzelman et al. conducted a prospective, randomized, controlled multi-center study on a total of 83 patients with type 1 or type 2 diabetes and a University of Texas degrade 1 or 2 diabetic foot ulcer ranging in size from 1 to 25 cm^2^. Among the 83 diabetic patients, 47 patients received a single application of the 4 cm × 4 cm Graftjacket and exhibited a complete healing rate of 69.6% and a mean healing time of 5.7 weeks at 12 weeks. The rest of the 39 patients received standard-of-care wound management consisting of moist-wound therapy with alginates, foams, hydrocolloids, or hydrogels at the discretion of the treating physician and had worse recovery effects, with a complete healing rate of 46.2% and a mean healing time of 6.8 weeks [99]. R.G. Frykberg found that compared with the effects of the human fibroblast-derived dermal substitute Dermagraft in 29 patients with non-healing diabetic foot ulcers, the treatment of 27 patients with MatriStem (MatriStem Wound Matrix and MatriStem MicroMatrix; ACell, Inc., Columbia, MD, USA), biologically derived dressings manufactured by decellularizing the inner lining of the porcine urinary bladder, led to a significantly improved quality of life and reduced costs [100].

Besides diabetic ulcers of the lower extremity, decellularized dermis exhibits its advantages in repairing defects of other tissues and organs. For instance, a comparison study of wound regeneration in injured sites of 10 patients showed that decellularized dermis treatment led to reduced fibrosis and a thicker regenerated dermis as compared with collagen-GAG scaffold treatment (Figure 1) [101]. When applied for staged breast reconstruction, acellular dermal matrix DermACELL (LifeNet Health, Virginia Beach, VA, USA), compared with non-acellular dermal matrix, induced less inflammation, fewer myofibroblasts, and decreased capsular contracture [102]. The bovine-derived acellular dermal matrix SurgiMend (TEI Biosciences, Boston, MA, USA) is also reported to be effective in breast reconstruction following skin-sparing mastectomy with reduced immunogenicity and foreign body responses, low implant loss, and high patient satisfaction [10]. Meso BioMatrix (DSM Biomedical, Exton, PA, USA), a novel porcine-derived acellular peritoneal matrix implant designed to reinforce soft tissue, has also been identified as a safe adjunct in breast reconstruction according to outcomes from a multi-center, prospective, single-arm trial of 25 patients [103]. The transplantation of acellular dermal matrix heterograft Heal-All (Zhenghai Biotechnology Co., Ltd., Yantai, China) in alveolar cleft patients demonstrated that acellular dermal matrix can increase the osteogenic effect of the bone graft and support bone formation in alveolar clefts [104]. A 15-year clinical study examined and compared the clinical outcomes of an acellular dermal matrix allograft and autogenous free gingival graft for gingival augmentation in patients presenting with the absence or deficiency of keratinized mucosa on homologous mandibular premolars (split mouth). The long-period observation of five male patients and seven female patients showed that both patients in the acellular dermal matrix allograft group and in the autogenous free gingival graft group had an increased keratinized tissue width and soft tissue thickness. Although a clinical parameter evaluation indicated that patients transplanted with autogenous free gingival grafts had advanced clinical outcomes, professional esthetic perception was better in patients transplanted with acellular dermal matrix allografts [111].

In addition to decellularized dermis, other acellular matrix has been applied in the treating skin wounds. Using the autologous extracellular matrix/stromal vascular fraction gel prepared from harvested fat, Chengliang Deng et al. applied an adipose-derived stem-cell-based cytotherapy, while avoiding the immune response caused by inadequate cell retention. A variety of different types of chronic wounds in 20 patients, including venous stasis ulcers, traumatic infections, diabetic ulcers, scar ulcers, and sarcoidosis, were treated with autologous extracellular matrix/stromal vascular fraction gels or a control negative pressure wound therapy. These chronic wounds that were originally unresponsive to standard wound care or skin grafting and were kept unhealed for more than 3 months. Grafting autologous extracellular matrix/stromal vascular fraction gels to these chronic wounds induced an average wound healing rate of around 35% per week, which was more than 3-fold the wound-healing rate in the control negative pressure wound therapy group. Reduced lymphocyte infiltration, elevated accumulation of collagen, and an increased amount of newly formed vessels were observed in the autologous extracellular matrix/stromal vascular fraction gel group based on hematoxylin-eosin staining, Masson’s trichrome staining, and immunohistochemistry labeling of the endothelial cell marker CD31. Accelerated wound healing and better histological outcomes demonstrated that the autologous extracellular matrix/stromal vascular fraction gel represents a promising treatment for chronic wounds [105]. Transgenic materials have also been put into use. Severe burns of a total of 50 patients were covered with a traditional dressing with vaslinlin-soaked gauze or acellular skin of α 1,2-fucosyltransferase and α-galactosidase double transgenic porcine or α-galactosidase single transgenic porcine. Alpha 1,2-fucosyltransferase and α-galactosidase transgenic modification reduced hyperacute xenograft rejection, while decellularization further diminished immune responses. Dressing burns with acellular skin xenografts obtained from transgenic porcine tissue reduced the perceived pain intensity at 7 days after surgery and condensed the length of the total hospitalization duration from 38 days in the traditional dressing control group to only 30 days [106].

### 5.2. Urogenital System Repair

Besides skin wounds, many other injured, damaged, or missing tissues and organs have also been replaced by acellular matrix. Acellular bladder matrix grafts generated from the decellularization of donor bladders have been used to treat urethral stricture diseases clinically. A total of 30 male patients having an anterior urethral stricture were randomly treated with acellular bladder matrix grafts or control buccal mucosal grafts in a 1:1 manner. A follow-up study showed that similar to patients treated with the favored material buccal mucosal, patients, especially patients with a healthy urethral bed, had a high recovery rate after acellular bladder matrix treatment, demonstrating the potential for the acellular bladder matrix in repairing the urethra [107].

### 5.3. Cardiovascular System Repair

The CorMatrix extracellular matrix (CorMatrix Cardiovascular Inc., Roswell, GA, USA), a collagen and glycosanmino glycans-enriched extracellular matrix derived from decellularized porcine small intestinal submucosa, has been used as a pericardial patch to repair post-myocardical infarction complications. The CorMatrix extracellular matrix repaired six patients with a left ventricle aneurysm, a patient with a ruptured left ventricle aneurysm, a patient with an ischemic ventricular septal defect and left ventricle aneurysm, and three patients with an ischemic ventricular septal defect with high stability under pressure, safety, and efficacy [108].

The decellularized porcine small intestinal submucosa also functions as an excellent biomaterial to treat other defects. The treatment of children with tympanic membrane perforation with SurgiSIS (Cook Surgical, Bloomington, IN, USA), a commercially available acellular tissue graft derived from the porcine small intestinal submucosa, led to stable tympanic membrane closures in 209 out of 217 children, having an even higher cure rate than autologous temporalis fascia repair [11].

A 1.8-year follow-up study of 106 patients with 118 decellularized allogeneic pulmonary artery patches of MatrACELL (Life-Net Health, Inc., Virginia Beach, VA, USA) demonstrated that there was no serious adverse effects, failure, or reoperation provoked by the decellularized patch materials. The decellularization procedure reduces residual antigens for the major histocompatibility complex, and thus, decellularized allogeneic pulmonary artery patches have obvious advantage over cryopreserved pulmonary homografts [109]. Pulmonary valve replacement with a total of 131 decellularized homografts reduced the rates of re-operation as compared with matched current gold-standard cryopreserved pulmonary homografts and bovine jugular vein conduits [110].

### 5.4. Nerve Defect Repair

In neural tissue engineering, acellular nerve grafts were constructed by collecting fresh peripheral nerves from traumatically amputated limbs and performing chemical extraction, decellularization, and irradiation. Decellularized nerve grafts obtained the natural structure of peripheral nerves and passed the evaluation of hepatitis B and hepatitis C, syphilis, the rapid plasma reagin test, and the toluidine red unheated serum test, and AIDS. Next, generated acellular nerve grafts were transplanted to bridge defects ranging from 1 to 5 cm in a total of 72 patients. Safety examinations showed that the transplantation of acellular nerve grafts did not induce abnormal wound responses or healing, local pain, allergic symptoms, or irregular laboratory test results. Functional assessments based on static 2-point discrimination and Semmes–Weinstein monofilament testing demonstrated that patients subjected to acellular nerve graft transplantation had a high excellent and good rate at 6 months after operation and an improved recovery over time. The determined superior clinical safety and efficacy characteristics of acellular nerve grafts imply that acellular nerve grafts can be used alternatives of the gold-standard autogenous nerve grafts to bridge nerve gaps [12].

Besides breast reconstruction, missing or deformed facial structures such as an external ear and nose can be reconstructed using decellularized materials. Reconstruction surgery of nine rhinoplasty patients with cross-linked decellularized animal caprine conchal cartilages achieved a very satisfied rate of 76% (seven out of nine patients) and a satisfied rate of 24% (two out of nine patients), while the reconstruction surgery of four rhinoplasty patients with autogenous cartilages achieved a 50% very satisfied rate and a 50% satisfied rate. Similar, reconstruction surgery of six microtia patients with cross-linked decellularized caprine conchal cartilages reached better outcomes as compared with autogenous cartilage grafting. These findings implied that cartilage reconstruction with decellularized cartilage tissues from animal sources is feasible [112].

## 6. Challenges and Future Prospects

In the past decade, dECM biomaterials have attracted great interest and research attention in the fields of tissue engineering and regenerative medicine. The therapeutic potential of these biomaterials (derived from cells, tissues, and organs) has been realized, as evidenced by the widespread publication of over 5000 articles. The Food and Drug Administration-approved transplants, including nerve conduits, skin, and ligaments, further demonstrate their clinical utility as tissue substitutes. Despite the successful applications of decellularized materials in the regeneration of various types of human tissues and organs, many clinical trials are in phase 1 studies with limited cases. Increased scales, as well as elongated follow-up studies, are needed to fully investigate the safety and effectiveness of decellularized materials.

One of the unresolved challenges in the clinical translation of decellularized scaffolds is to determine appropriate tissues and decellularization and sterilization methods, optimize the efficacy of these techniques, evaluate the quality of dECM, modify cell-free scaffolds with in vitro and in vivo studies, and address transplantation issues such as the timing of biomaterial transplantation. Select decellularization methods based on the type and characteristics of the organization. In addition to strengthening disinfection techniques, developing cell-removal technologies for reagents or products with lower toxicity would be beneficial. The ongoing research has improved the methods and approaches for developing non-immune biomaterials derived from cells and tissues with reproducible structures and biological functions. The versatility of dECM allows the development of various forms of biomaterials, such as powders [113], gel [114], sheets [115], 3D structures manufactured [116] using additives, and even tissues and organs with complete vascular structures and innervation. These non-immune and biologically active biomaterials and structures are superior to engineered matrices made from natural or synthetic biomaterials because they have natural structural and compositional properties, including binding motifs and biochemical signaling cues, playing a crucial role in host cell interactions and guiding tissue regeneration.

Although the above strategies are largely effective in the decellularization process, there are still many obstacles on the long road of dECM engineering. These include the need to improve cellular methods for effective product design, gain a deeper understanding of how dECM affects cell behavior, and achieve mechanical properties similar to natural tissues. At the same time, standardized sterilization and preservation methods, as well as characterization techniques, need to be developed to advance the clinical translation of decellularized tissue biomaterials and transplants.

## Figures and Tables

**Figure 1 bioengineering-13-00024-f001:**
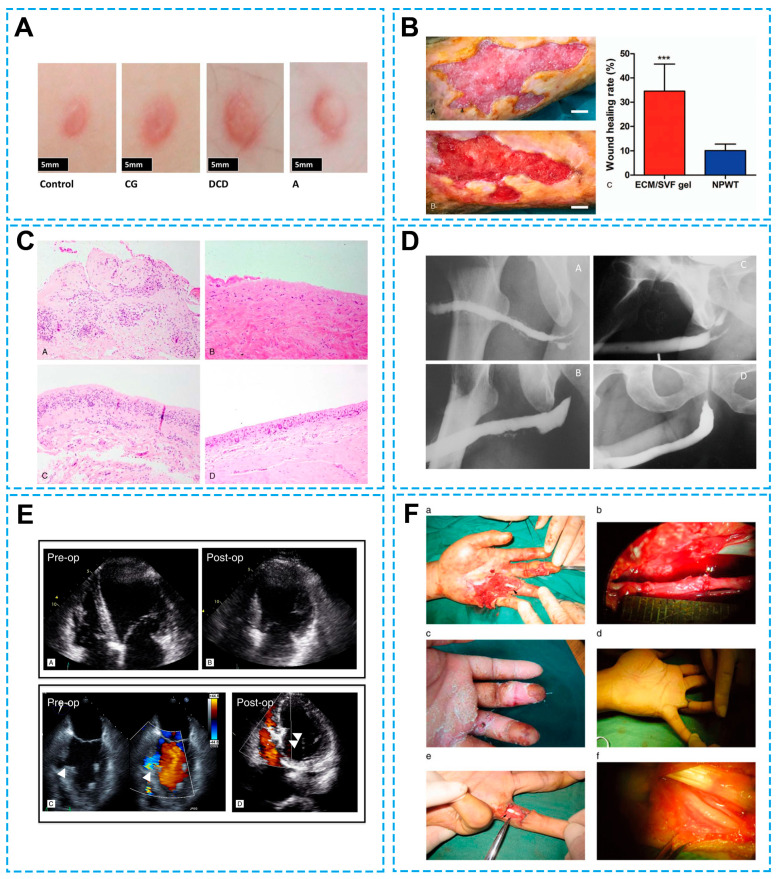
Clinical application outcomes of decellularized extracellular matrix biomaterials. (**A**) Decellularized dermis treatment reduced fibrosis with a thicker regenerated dermis as compared with the collagen-GAG scaffold treatment. CG: collagen scaffold, DCD: decellularized dermis, A: autograft [101] with permission of WILEY, © 2015. (**B**) Grafting autologous extracellular matrix/stromal vascular fraction gels to these chronic wounds induced an average wound healing rate of around 35% per week, which was more than 3-fold the wound-healing rate in the control negative pressure wound therapy group. (A) A chronic right leg wound with an original area of 19.65 cm^2^. (B) Residual area 14 days post-treatment with negative pressure wound therapy was 17.34 cm^2^. (C) Quantification of the wound area of the 2 treatment groups. *** *p* < 0.001.(Scale bar = 1 cm). ECM/SVF: extracellular matrix/stromal vascular fraction, NPWT: negative pressure wound therapy [105] with permission of LIPPINCOTT WILLIAMS & WILKINS, © 2018. (**C**) When applied for staged breast reconstruction, the acellular dermal matrix DermACELL, compared with a non-acellular dermal matrix, induced less inflammation, fewer myofibroblasts, and decreased capsular contracture, native (A,C), and acellular dermal matrix (B,D) [102], with permission of LIPPINCOTT WILLIAMS & WILKINS, © 2016. (**D**) Similar to patients treated with the favored material buccal mucosal, it had a high recovery rate after acellular bladder matrix treatment. (A,B) acellular bladder matrix; (C,D) buccal mucosa [107], with permission of LIPPINCOTT WILLIAMS & WILKINS, © 2008. (**E**) CorMatrix extracellular matrix has been used as a pericardial patch to repair post-myocardical infarction complications and with high stability under pressure, safety, and efficacy. Patient 2: anterolateral left ventricular aneurysm (A) and Dor repair (B). Patient 6: ischemic VSD (arrowhead, (C)) and patch VSD repair (D). ECM indicates extracellular matrix; VSD, ventricular septal defect [108], with permission of LIPPINCOTT WILLIAMS & WILKINS, © 2013. (**F**) Patients subjected to acellular nerve graft transplantation had a high excellent and good rate at 6 months after operation and an improved recovery over time. (a,b) Intraoperative photographs of nerve defects. (c) Two weeks after surgery. (d) Three months after surgery. (e,f) Intraoperative photograph of tendon and nerve detection [12], with permission of WILEY, © 2015.

**Table 1 bioengineering-13-00024-t001:** Core differences between organ-derived and cell-derived dECM.

Core Difference	Organ-Derived ECM	Cell-Derived ECM	Ref.
Sources	limited source of donor material	abundant sources	[15]
Immunogenicity risk	potential host immune response	greatly reduce the potential host response	[16]
Customizability of structure	structural fixation	prepared according to needs	[16]
Scale potential	NO	YES	[16]
Scalability	weak scalability	relatively flexible	[15,16]

**Table 2 bioengineering-13-00024-t002:** Physical, chemical, and biological methods commonly applied to decellularize tissues and cells.

Category	Treatment/Technique	Core Principle	Ref.
Physical	Freeze–thaw cycling	- Can consist of one or more freeze–thaw cycles - Lyses cell membrane via ice crystal formation	[26,29]
	Mechanical stirring	- Immerse the tissue in chemical reagents, detergents, and enzymes for mechanical stirring - Mechanical shock destroys cells in tissues	[27,30]
	Electroporation	- Forming micropores on the cell membrane leading to cell death	[31,32]
	Pressure	- High pressure disrupts cell membrane inside tissue	[33,34,35,36,37]
	Supercritical fluid	- Utilizes fluids (e.g., CO_2_) above critical temperature and pressure, which exhibit high diffusivity and low viscosity, enabling rapid penetration and efficient removal of cellular components while preserving ECM ultrastructure.	[38,39,40,41]
Chemical	Acids and bases	- Causes hydrolytic degradation of biomolecules and solubilizing cytoplasmic components	[42,43,44,45,46]
	Non-ionic detergents	- Destroying lipid–lipid or lipid–protein connections	[47,48,49,50,51,52,53]
	Ionic detergents	- Completely destroy the integrity of the cell membrane, dissolve the lipid bilayer structure	[54,55,56,57,58]
	Zwitterionic detergent	- Demonstrated a mild ability to maintain mechanical strength	[59,60]
	Hypotonic/hypertonic solutions	- Use the changes in osmotic pressure in the intracellular and extracellular environment to destroy the cell structure	[61]
	Alcohols	- Can dissolve non-polar substances: lipids	[27,54]
Biological methods	Trypsin	- Detaching cells from the tissue surface	[62]
	Nucleases	- Used to further remove the remaining DNA components	[63]
	Non-enzymatic agents	- Allowing cells to detach from the extracellular matrix	[64,65,66,67]

**Table 3 bioengineering-13-00024-t003:** List of decellularized extracellular matrix-related patents.

Decellularized Extracellular Matrix	Publication	Pub. No.	Pub. Date
Decellularized engineered tissues by growing cells on a substrate and decellularizing	United States Patent Application Publication	US2002/0115208 A1	2002/08/22
Decellularization method that contains enzymatic proteolytic digestion, nucleic acid removal, and cellular component removal	World Intellectual Property Organization, International Bureau	WO 2006/095342 A2	2006/09/14
Fabricated chondrocyte extracellular matrix from cartilage derived-chondrocytes by centrifuging and freeze-drying	World Intellectual Property Organization, International Bureau	WO 2008/126952 A1	2008/10/23
Decellularized extracellular matrix derived from the native or natural matrix of heart tissue	World Intellectual Property Organization, International Bureau	WO 2010/039823 A2	2010/04/08
Decellularized scaffolds with porous structures obtained by freeze-drying/lyophilizing bio-scaffolds and using detergent and alkaline solution	United States Patent Application Publication	US2012/0259415 A1	2012/10/11
Muscle implants comprising decellularized muscle matrices	World Intellectual Property Organization, International Bureau	WO 2014/008181 A2	2014/01/09
Decellularization using detergent perfusing, neutral buffer rinsing, and DNase solution delivery	United States Patent Application Publication	US2015/0238656 A1	2015/08/27
Decellularization by subjecting detergent-treated tissue samples to oscillation at a certain frequency	World Intellectual Property Organization, International Bureau	WO 2017/017474 A1	2017/02/02
Decellularization via non-thermal irreversible electroporation	United States Patent Application Publication	US2017/0209620 A1	2017/07/27
Bio-inks containing decellularized extracellular matrix	World Intellectual Property Organization, International Bureau	WO 2018/094166 A1	2018/05/24
Decellularization by closing afferent blood vessels to substantially seal a donor lobular organ with no common artery, eradicating blood, and perfusing detergent and enzymatic solutions	World Intellectual Property Organization, International Bureau	WO 2019/220091 A1	2019/11/21

**Table 4 bioengineering-13-00024-t004:** Patents of application and in granting procedure.

Decellularized Extracellular Matrix	Publication	Pub. No.	Pub. Date
Decellularized biphasic periodontal tissue grafts comprising interconnected polymer fiber scaffolds	World Intellectual Property Organization, International Bureau	WO 2016/049682 A1	2016/04/07
Decellularized extracellular matrix incorporated with a synthetic polymer	World Intellectual Property Organization, International Bureau	WO 2021/202974 A1	2021/10/07

**Table 5 bioengineering-13-00024-t005:** Clinical trials and market products of decellularized extracellular matrix biomaterials.

Category	Disease/Application Area	Product/Trial Name	Company/Institution	Outcome	References
Clinical Trial	Frey’s Syndrome	Acellular dermis (AlloDerm^®^)	LifeCell Corp.	Reduced gustatory sweating and complication rate	[98]
	Diabetic foot ulcer	Acellular dermal graft (Graftjacket^®^)	Wright Medical Technology	Complete wound closure in 12 out of 14 patients	[9,99]
	Non-healing diabetic foot ulcers	MatriStem^®^ Wound Matrix	ACell, Inc.	Improved quality of life and reduced cost	[100]
Clinical Trial	Wound regeneration	Decellularized dermis	-	Reduced fibrosis and thicker regenerated dermis	[101]
	Breast reconstruction	Acellular dermal matrix (DermACELL^®^)	LifeNet Health	Less inflammation, fewer myofibroblasts, and decreased capsular contracture	[102]
	Breast reconstruction	Bovine-derived acellular dermal matrix (SurgiMend^®^)	TEI Biosciences	Reduced immunogenicity, low implant loss, high patient satisfaction	[10]
	Breast reconstruction	Porcine-derived acellular peritoneal matrix (Meso BioMatrix^®^)	DSM Biomedical	Identified as a safe adjunct	[103]
	Alveolar cleft	Acellular dermal matrix heterograft (Heal-All^®^)	Zhenghai Biotechnology Co., Ltd.	Increased osteogenic effect and bone formation	[104]
	Chronic wounds	Autologous ECM/stromal vascular fraction gel	-	Increased wound healing rate, reduced lymphocyte infiltration, elevated collagen and vessel formation	[105]
	Severe burns	Acellular skin from transgenic porcine	-	Reduced pain intensity and shorter hospitalization	[106]
	Urethral stricture	Acellular bladder matrix grafts	-	High recovery rate, comparable to buccal mucosal grafts	[107]
	Post myocardial infarction complications	CorMatrix^®^ ECM (porcine SIS)	CorMatrix Cardiovascular Inc.	High stability, safety, and efficacy as a pericardial patch	[108]
	Tympanic membrane perforation	Porcine SIS (SurgiSIS^®^)	Cook Surgical	Stable tympanic membrane closures in 209/217 children	[11]
	Pulmonary valve replacement	Decellularized allogeneic pulmonary artery patches (MatrACELL^®^)	LifeNet Health	No serious adverse effects, failure, or reoperation	[109]
	Pulmonary valve replacement	Decellularized homografts	-	Reduced re-operation rates	[110]
	Nerve defects	Acellular nerve grafts	-	A high excellent and good rate	[12]
Market Product	Wound healing	Biodesign^®^ (SIS)	Evergen (formerly Cook Biotech, Alachua, FL, USA)	Commercially available for soft tissue repair and wound management	-
	Wound healing	OASIS^®^ (SIS)	Evergen (formerly Cook Biotech, Alachua, FL, USA)	Commercially available for acute and chronic wound treatment	-
	Cardiovascular repair	Decellularized pulmonary heart valve	Artivion (formerly Cryolife, Kennesaw, GA, USA)	Marketed since the late 2000s for pulmonary valve replacement	-

## Data Availability

No new data were created in this review article.

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
