# Peer review of "From Production to the Clinic: Decellularized Extracellular Matrix as a Biomaterial for Tissue Engineering and Regenerative Medicine"

_bioengineering, 2025, doi:10.3390/bioengineering13010024_

Round 1
Reviewer 1 Report
Comments and Suggestions for Authors
This review outlines methods for preparing and evaluating decellularized extracellular matrix as a biomaterial for tissue engineering and regenerative medicine, and also summarizes patent information and clinical trial results. I felt that the first half of the article and the description of the physical and chemical methods used to produce the decellularized extracellular matrix were fine, but I noticed some issues with the biological methods, as described below. The physical and chemical methods are somewhat outside my area of ​​expertise, so I am concerned that similar issues may exist. 
The descriptions of patents and clinical trials in the latter half of the manuscript are lengthy and seem to waste pages. Since there are several tables, it would be better to make the descriptions more concise and easier to read, but I would like to point out that this is acceptable as long as it conforms to the journal's style.
This manuscript contains tables but no figures. To attract readers' attention, I recommend providing a diagram showing the process from decellularized extracellular matrix production to clinical application.
Comments
Page 9, line 349: Is ref. 73 really a reference for Enzymatic technique for decellularization?
Page 9, line 350: Nuclease, lipase, heparinase, and hyalronidase are not proteolytic enzymes.
Page 10, line 359: Why is cohesin involved in cell-matrix detachment even though it is an intracellular protein? Authors should provide a reference.
Page 10, line 373: Therefore, it is widely used to digest lipase….. ---> Lipase is used to digest lipase.
Page 10, line 374: But it will reduce the content of glycosaminoglycans to a certain extent. ---> Authors should provide a reference.
Page 10, line 383: Is there a term for "trypsin detergents"?
Page 10, line 387: Rhosponin B → Not found by search. According to ref 67, it is latrunculin B.
Page 10, line 388: Rhospongin B → Not found by search. According to ref 67, it is latrunculin B.
Author Response
- Page 9, line 349: Is ref. 73 really a reference for Enzymatic technique for decellularization?
Response to Reviewer: Thank you for pointing this out. We have replaced Ref. 73 with more appropriate references on enzymatic decellularization.
- Page 9, line 350: Nuclease, lipase, heparinase, and hyalronidase are not proteolytic enzymes.
Response to Reviewer: Thank you for pointing this out. We have revised the text accordingly to clarify enzyme categories in lines 389-391.
- Page 10, line 359: Why is cohesin involved in cell-matrix detachment even though it is an intracellular protein? Authors should provide a reference.
Response to Reviewer: We apologize for the incorrect terminology. “Cohesin” was a misstatement; we intended to refer to proteins involved in cell–matrix adhesion. We have corrected the sentence. The sentence now reads: “trypsin can selectively cleave peptide bonds at the carboxy-terminus of arginine and lysine in adhesion proteins, thereby detaching cells from the tissue surface”in lines 398-400.
- Page 10, line 373: Therefore, it is widely used to digest lipase….. ---> Lipase is used to digest lipase.
Response to Reviewer: We thank the reviewer for catching this redundancy. The sentence has been corrected to clarify that lipase is used to hydrolyze lipid components in tissues in lines 413-415.
- Page 10, line 374: But it will reduce the content of glycosaminoglycans to a certain extent. ---> Authors should provide a reference.
Response to Reviewer: A relevant citation has been inserted in line 415.
- Page 10, line 383: Is there a term for "trypsin detergents"?
Response to Reviewer: We agree that “trypsin detergents” is not a standard term. We have revised the wording to more accurately describe the combined use of trypsin with detergents. Changed to: “Ethylenediaminetetraacetic acid and ethylene glycol tetraacetic acid can be used in combination with trypsin and detergents…”in line 423.
- Page 10, line 387: Rhosponin B → Not found by search. According to ref 67, it is latrunculin B.
Response to Reviewer: We thank the reviewer for this correction. All instances of “Rhosponin B”have been replaced with “Latrunculin B.”
- Page 10, line 388: Rhospongin B → Not found by search. According to ref 67, it is latrunculin B.
Response to Reviewer: We thank the reviewer for this correction. All instances of “Rhosponin B”have been replaced with “Latrunculin B.”
Reviewer 2 Report
Comments and Suggestions for Authors
The authors are thanked for preparing this review article. This reviewer suggests limiting the article to a more manageable scope to improve the impact of the paper. In trying to cover so many different topics, so broadly, it is not possible to provide complete detail an any one area.
Author Response
Response to Reviewer: Thank you very much for taking the time to review our manuscript and providing valuable feedback on its structure and content. We fully understand and agree with your point that the article covers a wide range and suggests focusing on enhancing the impact of the paper. Exploring a specific topic in depth within a limited space can indeed help improve the academic depth of the article.
When writing this review, our original intention was to provide readers with a systematic overview of the preparation, characterization, and clinical translation of "extracellular matrix", especially for researchers who are new to the field or need a cross disciplinary understanding. Therefore, we intentionally incorporated multiple aspects such as sources, methods, representations, patents, and clinical applications when organizing content, in order to present the overall development trajectory and transformation potential of the field.
We understand your concern about the depth of the content, so we have tried our best to select representative studies, key methods, and typical cases in each chapter for elaboration, and pointed out the current challenges and future directions in the discussion section. We believe that this' breadth first 'review still has certain academic value and reference significance at the current stage.
In summary, we hope to further strengthen the logical connection between each part while retaining the existing structure, and clarify the positioning and applicable readership of this review in the text to help readers better understand the writing intention and structural arrangement of this article.
Thank you once again sincerely for your constructive feedback, which has been very helpful for us to improve the expression and positioning of the article.
Reviewer 3 Report
Comments and Suggestions for Authors
This review summarizes the state of the art in ECM decellularization. While it tries to provide as much information as possible and describe all techniques and tissues it looses focus and is is partially hard to read because some paragraphs an chapters read like a list of 1-2 sentences summaries of published work. It was really hard for me to read especially chapter 4. What I miss is a second level of interpretation. For example, diagrams and graphs can be added that show different techniques, explain there pros and cons and help the reader to find the right path of decellularization for a specific tissue and application. This could provide some added value to this review. Also obstacles and limitations could be expanded.
Specific points
Table 2: “Supercritical fluid -Simpler and more efficientSupercritical fluid -Simpler and more efficient.” This sentence does not explain the core principle as indicated.
Authors should pls not use phrases that somehow describe a process without any details such as “Different studies use different methods according to different target tissues, from the 167 selection of reagents, the order of use, the concentration, the strength, speed and time of 168 stirring. Even for the same tissue, different scholars will adopt different decellularization 169 methods according to the difference in research purpose and material application“ – this sentence (example) does not say anything.
“For thinner tissues, such as small intestinal mucosa, the stirring time can be shortened; 171 for thicker tissues, such as esophageal tissue, the stirring intensity and speed should be 172 increased while the stirring time is extended.” – for sure, but this again lacks some in depth discussion.
Chapter 4 is particular boring to read and there is not much of structure. Example; After periodontal tissue some sentence about muscle and next some diffuse part about lobular organ.
The overall language of this article is good, but there are some typos and some sentences read very strange. Example: A high very satisfied rate and satisfied rate. Pls add another round of proofreading.
“In conclusion, with regard to the types and concentrations of chemicals used in the 343 decellularization process, it is generally more beneficial to use different chemicals and 344 form an appropriate combination to exert the best decellularization efficiency.” --- is it? There is no data discussed. If this is a hypothesis, indicate it is one and explain your arguments. What is a good combination of chemicals? What makes sense to be tested and how to decide which chemicals.
The overall language of this article is good, but there are some typos and some sentences read very strange. Example: A high very satisfied rate and satisfied rate. Pls add another round of proofreading.
Comments on the Quality of English LanguageOverall okay. Quite some typos but editorial process will solve it.
Author Response
This review summarizes the state of the art in ECM decellularization. While it tries to provide as much information as possible and describe all techniques and tissues it looses focus and is is partially hard to read because some paragraphs an chapters read like a list of 1-2 sentences summaries of published work. It was really hard for me to read especially chapter 4. What I miss is a second level of interpretation. For example, diagrams and graphs can be added that show different techniques, explain there pros and cons and help the reader to find the right path of decellularization for a specific tissue and application. This could provide some added value to this review. Also obstacles and limitations could be expanded.
Response to Reviewer: We sincerely thank the reviewer for the thorough and constructive feedback. We added a new graph abstract and highlight to help readers better understand the content and focus of the article.
Specific points
- Table 2: “Supercritical fluid -Simpler and more efficientSupercritical fluid -Simpler and more efficient.” This sentence does not explain the core principle as indicated.
Response to Reviewer: We thank the reviewer for noting this incomplete description. We revised the entry in Table 2 to clearly state the core principle.
- Authors should pls not use phrases that somehow describe a process without any details such as “Different studies use different methods according to different target tissues, from the 167 selection of reagents, the order of use, the concentration, the strength, speed and time of 168 stirring. Even for the same tissue, different scholars will adopt different decellularization 169 methods according to the difference in research purpose and material application“ – this sentence (example) does not say anything.
Response to Reviewer: We agree that such sentences are uninformative. The original statement lacked specific information, and in lines 174-182 we have replaced it with a deep analysis based on research cases
- “For thinner tissues, such as small intestinal mucosa, the stirring time can be shortened; 171 for thicker tissues, such as esophageal tissue, the stirring intensity and speed should be 172 increased while the stirring time is extended.” – for sure, but this again lacks some in depth discussion.
Response to Reviewer: In lines 182-193 We expand this point with specific examples and rationale.
- Chapter 4 is particular boring to read and there is not much of structure. Example; After periodontal tissue some sentence about muscle and next some diffuse part about lobular organ.
Response to Reviewer: We acknowledge that Section 4 is overly descriptive and list-like. We will restructure it to improve flow.
- “In conclusion, with regard to the types and concentrations of chemicals used in the 343 decellularization process, it is generally more beneficial to use different chemicals and 344 form an appropriate combination to exert the best decellularization efficiency.” --- is it? There is no data discussed. If this is a hypothesis, indicate it is one and explain your arguments. What is a good combination of chemicals? What makes sense to be tested and how to decide which chemicals.
Response to Reviewer: In lines 363-378 we revised this statement to clarify that it is a consensus observation from the literature and provide examples of effective combinations.
- The overall language of this article is good, but there are some typos and some sentences read very strange. Example: A high very satisfied rate and satisfied rate. Pls add another round of proofreading.
Response to Reviewer: We thank the reviewer for the positive note on language. We will perform a thorough proofreading of the entire manuscript to correct typos and improve sentence fluency.
Round 2
Reviewer 1 Report
Comments and Suggestions for Authors
The authors made changes in response to the reviewers' comments, and the manuscript has been improved. I have no further suggestions or comments.
Author Response
comment1:The authors made changes in response to the reviewers' comments, and the manuscript has been improved. I have no further suggestions or comments.
Response to Reviewer: We would like to express our sincere gratitude for the valuable comments and constructive suggestions provided by you, which have greatly helped us improve the quality of our manuscript.
Reviewer 2 Report
Comments and Suggestions for Authors
Please change table 5 to include both clinical trails and market products. At minimum, the authors should add some of the SIS products made Evergen (formerly Cook Biotech). A couple examples are Biodesign and OASIS. They should also add the decellularized pulmonary heart valve from Artivion (formerly Cryolife). This has been on the market since the late 2000s.
Author Response
Comment2: Please change table 5 to include both clinical trails and market products. At minimum, the authors should add some of the SIS products made Evergen (formerly Cook Biotech). A couple examples are Biodesign and OASIS. They should also add the decellularized pulmonary heart valve from Artivion (formerly Cryolife). This has been on the market since the late 2000s.
Response to Reviewer:
Thank you very much for carefully reviewing our manuscript and providing valuable revision suggestions. We fully agree with your suggestion that information on commercially available decellularized matrix products should be added to Table 5 to more comprehensively reflect the current clinical translational status in this field.
Based on your suggestion, we have made the following modifications and extensions to Table 5:
On the basis of the original clinical trial information, a new column called "Category" has been added to distinguish between "Clinical Trials" and "Market Products". Add columns for "Product/Test Name" and "Company/Institution" to clearly present the product name and its developer.
Added the key commercial products you mentioned:
Biodesign ® With OASIS ® (All are small intestinal submucosal matrix SIS products from Evergen, formerly Cook Biotech), and are classified in the field of "wound healing". Decellularized pulmonary artery valve (from Artivion, formerly Cryolife), classified as "cardiovascular repair" and noted to have been on the market since the late 2000s.
The revised table now covers clinical trials and marketed products, better reflecting the complete path of dECM from research to clinical translation and commercialization.
We believe that this supplement significantly enhances the information content and representativeness of the table. At the same time, we improved the language throughout the article to express the research more clearly. Thank you again for your highly constructive feedback.
Reviewer 3 Report
Comments and Suggestions for Authors
The new version of the manuscript has been improved, in particular the graphical abstract is well structured and provides the reader with a good overview. I have no further suggestions.
Author Response
comment3: The new version of the manuscript has been improved, in particular the graphical abstract is well structured and provides the reader with a good overview. I have no further suggestions.
Response to Reviewer: We would like to express our sincere gratitude for the valuable comments and constructive suggestions provided by you, which have greatly helped us improve the quality of our manuscript.